# Predicting and Modeling Wildfire Propagation Areas with BAT and Maximum-State PageRank

**Wei-Chang Yeh *** and **Chia-Chen Kuo**

Integration and Collaboration Laboratory, Department of Industrial Engineering and Engineering Management, National Tsing Hua University, Hsinchu City 300, Taiwan; cckuo@ie.nthu.edu.tw
* Correspondence: yeh@ieee.org; Tel.: +886-(3)574-2443



**Featured Application: The Climate conditions as factors in the equations and random values will be incorporated into the featured application for future research model.**

**Abstract:** The nature and characteristics of free-burning wildland fires have significant economic, safety, and environmental impacts. Additionally, the increase in global warming has led to an increase in the number and severity of wildfires. Hence, there is an increasing need for accurately calculating the probability of wildfire propagation in certain areas. In this study, we firstly demonstrate that the landscapes of wildfire propagation can be represented as a scale-free network, where the wildfire is modeled as the scale-free network whose degree follows the power law. By establishing the state-related concepts and modifying the Binary-Addition-Tree (BAT) together with the PageRank, we propose a new methodology to serve as a reliable tool in predicting the probability of wildfire propagation in certain areas. Furthermore, we demonstrate that the proposed maximum-state PageRank used in the methodology can be implemented separately as a fast, simple, and effective tool in identifying the areas that require immediate protection. The proposed methodology and maximum-state PageRank are validated in the example generated from the Barabási-Albert model in the study.

**Keywords:** wildfire; fire model; model performance; PageRank; binary-addition tree (BAT); states; maximum-state; scale-free network; Barabási-Albert model

## 1. Introduction

The network research is very popular and has been applied to research in many fields [1–38]. Recently, wildfires have seen unbelievable growth in size, frequency, and complexity. As defined in [28], wildfire is a non-structure and non-prescribed that occurs in wildlands. Wildfires often ignite close to wildland-human boundaries, resulting in severe damage to assets [28–30], air pollution [31,32], loss of plants and forest [33–35], and high fatalities [36,37]. With global warming, a 1° increase in the climatic temperatures causes a 15% growth in wildfires [38]; the wildfires in turn increase the temperature, thereby further increasing the number of wildfires, forming an infinite loop [28,38].

Wildfire is a severe practical problem [28,38], and does not always occur in the same area, such as a town or city, but expands to the neighboring areas, and is affected by weather conditions, e.g., wind, rain, etc., and location surface, such as forest or lake [36]. Hence, unlike the traditional path problem [39], which states that all events occur in a path, and the spanning tree problem [40], which has fixed destinations, the wildland fire problem requires a new algorithm to estimate its occurrence and the possible results of its propagation [32–34].

Several deterministic and probabilistic models have been proposed to predict the wildfire propagation from node-to-node (cell/city/area) [28–38]. Among them, graph-based methodologies,

including the path-based and tree-based methodologies, are popular [32–34]. Furthermore, many probabilistic wildfire propagation models have been proposed based on the continuous-time Markov chain [41].

The path-based algorithms [32–34] focus on identifying the deterministic or stochastic paths with the shortest distance, minimal travel time, or high probability. The tree-based algorithms emphasize on one-source to many-targets and one ignition to many different and known nodes, rather than the one-source to one-target, i.e., from ignition to another target, which is the basis of path-based algorithms [39,40]. The tree-based algorithms include all cellular automata-related algorithms. Furthermore, the path-based and tree-based algorithms require the final destination information [32–34,39,40].

Free-burning wildfire propagation exhibits extremely inconstant and chaotic behaviors [32,35,41]. In this study, we demonstrate that the wildfire propagations can be modeled as a scale-free network [42,43], which is a common and effective method to depict complex systems. To the best of our knowledge, this is the first time that the wildfire propagations are modeled as scale-free networks among the existing literature.

Many practical and important networks are scale-free networks where the node degree [42–45], i.e., the number of edges connected to a node, follows an asymptotical power-law distribution [44]. A model of the scale-free network is a scale-free model [42]. Several examples of the study and use of scale-free networks are available and include the sharing of information among nodes in the network via links, such as the world wide web, protein-interaction networks, metabolic networks for eukaryotes and bacteria, co-authorship/collaborative network, and power grid of the western United States [42–45].

Among the different models of scale-free [45] networks, the Barabási-Albert model [45] is the most popular with numerous variations and generalizations [45]. Hence, the Barabási-Albert model is used to model the propagation characteristic property of the wildland fire and generate test data to validate the performance of the proposed algorithm.

The Binary-Addition-Tree (BAT) proposed by Yeh [46] is similar to the depth-first-search (DFS) [22,47,48], breadth-first-search (BFS) [49–51], universal generating function methodology (UGFM) [22,43–53], etc. However, BAT is easier to understand and convenient to code. The most important advantage of BAT is its flexible unique search procedure, and its valuable and easy implementation [46]. Hence, it has been adapted to solve the proposed wildfire propagation.

This study aims to calculate the probability of the wildfire propagation areas and quickly determine the areas that require immediate protection. Using the proposed novel state concepts and traditional PageRank, whose details are provided in Section 2.3, this study proposes an innovative methodology based on BAT [46] and PageRank [54,55] to predict the probable areas affected by the wildfire propagation that is modeled using the scale-free network generated from the Barabási-Albert model [45] and assess the risk of wildfire spread. Furthermore, the proposed maximum-state PageRank is implemented to quickly identify the node that requires immediate protection.

The rest of this study is organized as follows. An overview of the wildfire propagation models, scale-free networks [42–45], Barabási-Albert model [45], and BAT [46] are presented in Section 2. The pseudo-code, details of the BAT in solving the given problem, and discussion regarding the novelty of the proposed BAT, innovative states, state PageRank, state probability, and maximum-state PageRank are presented in Section 3. An experiment is presented in Section 4 to demonstrate the applicability of the proposed BAT. Furthermore, the analytical results are reported in this section. Section 5 concludes and summarizes the discussion.

## 2. Overview of Wildfire Propagation, Scale-Free Network, Barabási-Albert Model, and BAT

Viegas [56] indicated that there are many factors involved in the wildfire, and there are no general models yet available to perform this task with a sufficient degree of accuracy and reliability. There is a wide range of fire propagation models, ranging from the empirical to the physical [57,58]. The wildfire

propagation area estimation problem was modeled as a scale-free network using the Barabási-Albert model [45] and solved by modifying the BAT [46]. Therefore, before deliberating the proposed BAT, a general background of the wildfire propagation [28,38], scale-free network [15–45], Barabási-Albert model [45], and BAT [46] are presented in this section.

### 2.1. Wildfire Propagation

The landscape is mapped into an undirected binary-state network $G(V,E)$, where $V$ is the set of nodes representing towns, cities, forests, etc., and $E$ is the set of undirected links connecting the nodes in $V$, such that two neighboring nodes are connected by an undirected link. Each node exhibits two states: with or without wildfire, and each link is assumed to be always perfect without any possible failure.

For example, Figure 1 illustrates a landscape network representing eight areas labeled 0–7 with 15 links connecting these nodes, e.g., nodes 3 and 7 are neighboring areas as there is a link between the two nodes.

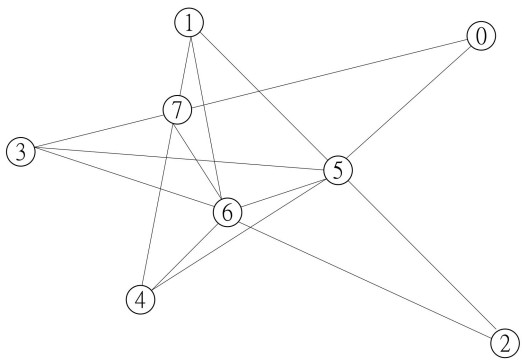

**Figure 1.** Example network.

The scale-free network grows independent of the number of nodes, i.e., the size of the network and underlying structure remains the same. The propagation of wildfire satisfies this property, and is thus considered as a scale-free model in this study. In this subsection, the model of the wildfire propagation is further modeled as a scale-free model with three prospects: ignition and spread of wildfire, probability of burning, and fire head intensity.

The ignition and spread of wildfire exhibit a high correlation with heat diffusion, which is a general observed natural phenomenon [59] and can be studied as a scale-free network. Heat diffusion is observed from high-temperature regions to lower temperature regions and this phenomenon is observed in the propagation of wildfire. Therefore, recently, heat diffusion has been implemented to model numerous networks whose information or flow is shared and propagated on online social networks.

From the MedSpread model, the probability of burning is defined in [31] as

$$P = (1 - e^{-r})p \tag{1}$$

where $r$ is the spread rate and $p$ is the fire intensity, representing the relation between the fire spread and the probability of burning. Equation (1) is similar to the power functions.

The fire head intensity $q$ (kW/m) is the rate of energy release per unit length of the fire head (kW/m). Regardless of the depth of the fire and fuel characteristics, $q$ is calculated as follows [36]:

$$q = 260 \, L^{2.174} \tag{2}$$

where $L$ is the flame length (m). Equation (2) is a power function.

Any model that follows the power law can be considered as a scale-free model, as reported in [44]. From the aforementioned heat diffusion and Equations (1) and (2), the wildfire propagation can be

modeled as a scale-fee network. Hence, we adopt the scale-free model to estimate the possible affected areas during wildfire propagation.

### 2.2. Scale-Free and Barabási-Albert Model

In scale-free networks [42–45], the node degree follows a power-law distribution [44], such that the network size is independent of the scale. Hence, the node degree is the core of a scale-free network. Two different nodes are observed in scale-free networks: hub nodes and non-hub nodes, wherein the degrees of the nodes are greater than and less than a certain threshold, respectively [44].

The scale-free networks adopted here are generated from a minimal model called the Barabási-Albert model proposed by Barabási and Albert in 1999 [45]. Inspired by the observation that World Wide Web (WWW) exhibits power-law (or scale-free) degree distributions, the Barabási-Albert model [45] was proposed as the first scale-free model in developing the network based on the following two steps:

STEP B1.  Growth: Nodes are individually added into the network such that the network size, i.e., the number of nodes, increases over time.

STEP B2.  Preferential attachment: The probability of the newly added node being connected with an old node is dependent on the degree of the old node.

Growth and preferential attachment are extremely common in existing real-life networks. Growth indicates that the network grows from time to time by adding new nodes, and preferential attachment describes the growth of the network, i.e., the connection between new nodes and old nodes [45]. Furthermore, STEP B2 indicates that the degree of nodes follows the power law and preferential attachment is a probabilistic mechanism [45].

We assume that the number of nodes is $n$: 0, 1, 2 ..., $(n-1)$, $\text{Deg}(i)$ is the degree of node $i$, and $\text{Pr}(i)$ is the probability that a new node is connected to node $i$. STEP B2 is based on $\text{Pr}(i)$ to predict the probability of the new node being added to the network in connection with node $i$. $\text{Pr}(i)$ can be defined as

$$\text{Pr}(i) = \frac{\text{Deg}(i)}{\sum\limits_{j=0}^{n-1} \text{Deg}(j)} \tag{3}$$

It can be seen from Equation (3) that in the scale-free network, most of the nodes are non-hub nodes with a few links. Based on Step B2, a few high-degree non-hub nodes can gradually transform into hubs because new nodes are more likely to connect with them [45]. This is called a rich-get-richer phenomenon [42–45]. In other words, the number of edges among all nodes approximately follows the power law property [44].

Table 1 presents the adjacency matrix [60] that can be used to determine the degree of each node in Figure 1, which is used as an input to the proposed method.

**Table 1.** Adjacency matrix for Figure 1.

| $i$ | 0 | 1 | 2 | 3 | 4 | 5 | 6 | 7 |
|---|---|---|---|---|---|---|---|---|
| 0 | | | | | | 1 | | 1 |
| 1 | | | | | | 1 | 1 | 1 |
| 2 | | | | | | 1 | 1 | |
| 3 | | | | | | 1 | 1 | 1 |
| 4 | | | | | | 1 | 1 | 1 |
| 5 | 1 | 1 | 1 | 1 | 1 | | 1 | |
| 6 | | 1 | 1 | 1 | 1 | 1 | | 1 |
| 7 | 1 | 1 | | 1 | 1 | | 1 | |
| Deg($i$) | 2 | 3 | 2 | 3 | 3 | 6 | 6 | 5 |

The normalized adjacency matrix presented in Table 2 is calculated by dividing the element values in Table 1 by their degrees. The normalized adjacency matrix is required to calculate the PageRank values discussed in the next section. The original concept of BAT is first proposed and used in Table 2 of [61] without naming it.

**Table 2.** Normalized adjacency matrix based on each column.

|   | 0 | 1 | 2 | 3 | 4 | 5 | 6 | 7 |
|---|---|---|---|---|---|---|---|---|
| 0 | 0 | 0 | 0 | 0 | 0 | 0.166667 | 0 | 0.2 |
| 1 | 0 | 0 | 0 | 0 | 0 | 0.166667 | 0.166667 | 0.2 |
| 2 | 0 | 0 | 0 | 0 | 0 | 0.166667 | 0.166667 | 0 |
| 3 | 0 | 0 | 0 | 0 | 0 | 0.166667 | 0.166667 | 0.2 |
| 4 | 0 | 0 | 0 | 0 | 0 | 0.166667 | 0.166667 | 0.2 |
| 5 | 0.5 | 0.333333 | 0.5 | 0.333333 | 0.333333 | 0 | 0.166667 | 0 |
| 6 | 0 | 0.333333 | 0.5 | 0.333333 | 0.333333 | 0.166667 | 0 | 0.2 |
| 7 | 0.5 | 0.333333 | 0 | 0.333333 | 0.333333 | 0 | 0.166667 | 0 |

### 2.3. PageRank

In recent years, numerous algorithms have been proposed to rank the importance of nodes in scale-free networks. Among these algorithms, PageRank used by Google in their search engine was historically the first and most well-known algorithm [54,55]. Simply follows the basic property of a scale-free network, whereas PageRank assumes that many important nodes (websites) are expected to receive more links from other nodes (websites) [54].

The PageRank of a node (page/website) $i$ after obtaining a degree, as listed in Table 1, is given as follows [44]:

$$PR(i) = (1 - d) + d \times [PR(i_1)/Deg_{out}(i_1) + \ldots + PR(i_n)/Deg_{out}(i_n)] \tag{4}$$

where

- $i_1, i_2, \ldots, i_{ni}$ are nodes with links to node $i$
- damping factor $d$ is a number between 0 and 1 and it accounts for the $(1 - d)$ chance of jumping into an arbitrary node.
- $Deg_{out}(j)$ is the out-degree number of a node, i.e., the number of links going from node $j$.

As an important node refers to other important nodes that are linked to it, the steps of the PageRank algorithm [53] are listed in Algorithm 1.

---

**Algorithm 1** PageRank Algorithm [53]

---

**Input** A scale-free network $G(V, E)$ with node set $V = \{0, 1, \ldots, (n - 1)\}$ and link set $E$.
**Output** Probability distribution, i.e., the PageRank value, PR($i$) for all nodes $i \in V$.
1: Let time $t = 0$. Set the initial PR*(i)* as $1/n$ for node $i$ and $i = 0, 1, \ldots, (n - 1)$.
2: Update PR($i$) based on Equation (4) for all i $\in$ V.
3: Redistribute PR*(i)* equally among the other nodes for all nodes $i$ with Degout($i$) = 0.
4: **if** there is no change in PR*(i)* for $i = 0, 1, 2, \ldots, (n - 1)$ **then go to** Stop.
5: **else if** let $t = t + 1$ **return** to line 2.

---

Line 1 initializes all node rankings with equal weights. Line 2 implements the basic concept of the PageRank defined in Equation (4) to update the node ranking recursively by adding the weight of every node that is linked to it and dividing it by the number of links emanating from the referring node [54]. The redistribution in line 3 is based on an underlying assumption that users will continue to search WWW if they enter a node (webpage in [54]) without exiting the link, i.e., a dead end. The procedure is repeated to update the rough PageRank value of each node in line 3 to converge to a constant number.

The output value PR(*i*) for all nodes *i* in the PageRank algorithm indicates the probability distribution of a user clicking on webpages (nodes) randomly to arrive at any particular webpage (node) [54,55]. Hence, the PR(*i*) can be adapted to estimate the probability that a wildfire randomly ignites and propagates in an area. Thus, to model the growth of wildfire propagation and describe the related burning areas, PageRank is adopted to measure the importance of nodes (areas) and is then transformed to the appropriate probability, which is used in the proposed dynamic BAT.

*2.4. BAT*

BAT was first proposed by Yeh [46] in 2020. It is a simple tree based on binary addition, and is used to generate all the possible binary-state vectors whose values of state coordinates are either 0 or 1. Experimental results have confirmed that BAT is better than the conventional Inclusion-Exclusion method, which is based on depth-first-search (DFS) [22,47], and the conventional Inclusion-Exclusion method is better than the Sum-of-disjoint product method, which is based on the breadth-first-search (BFS) [22,23,48,49]. Hence, BAT outperforms BFS and DFS in terms of the required computer memory and efficiency [46].

The flexibility of BAT highlights the malleability of its searchability and the ease of modification and customization with various practical applications [46]. Therefore, we customized it for the problems considered in this study.

Let us assume that $Y_k$ is the *k*th obtained vector with *m* coordinates and the *i*th coordinate $Y_k(i)$ is the state of link $a_i \in E$ in $Y_k$. The original BAT is link-based and all coordinates in its calculated vectors are the states of the links. In the original link-based BAT, $Y_{k+1}$ is generated after adding 1 to $Y_k$ by treating $Y_k$ as a binary number whose $i^{th}$ digit is equal to $Y_k(i)$. The overall pseudocode of BAT is shown in Algorithm 2.

---

**Algorithm 2** BAT Algorithm [46]

---

**Input** A binary-state network *G(V, E)* with two states, either 0 or 1, to each link and $V = \{0, 1, 2, \ldots, n-1\}$
**Output** All possible non-duplicate link-based state vectors.
1: Let SUM = 0, vector index *k* = 1, and *Y1* = *Y* be a zero vector with *m* coordinates which represented the states of the related links.
2: Let coordinate index *i* = *m*.
3: **If** *Y(i)* = 0, let *Y(i)* = 1, *k* = *k* + 1, $Y_k$ = *Y*, SUM = SUM + 1 **then go to** line 5.
4: Let *Y(i)* = 0. **If** *i* > 1, let *i* = *i* − 1 **then go to** line 3.
5: **If** SUM = m, halt, and $Y_1, Y_2, \ldots, Y_k$ are all possible state vectors **return** line 2.

---

Algorithm 2 begins with a zero vector, as shown in line 1. All the other vectors are generated by adding 1 in accordance with the binary addition in sequence in the form of a loop from the line 2 to the line 4. The algorithm stops when SUM = *m*, i.e., all the coordinates are 1, in line 5 [46].

Considering *m* = 5, we have the zero vector $Y_1 = (0, 0, 0, 0, 0)$ whose binary number is 00000. Using the aforementioned pseudocode from the line 2 to the line 4., the first five vectors are easily obtained and represented in binary numbers as follows:

$$00000 + 1 = 00001, \tag{5}$$

$$00001 + 1 = 00010, \tag{6}$$

$$00001 + 1 = 00011, \tag{7}$$

$$00011 + 1 = 00100. \tag{8}$$

It can be seen from Equations (5)–(8), $Y_2 = (0, 0, 0, 0, 1)$, $Y_3 = (0, 0, 0, 1, 0)$, $Y_4 = (0, 0, 0, 1, 1)$, and $Y_5 = (0, 0, 1, 0, 0)$. Similarly, all the vectors obtained in BAT are presented in Table 3, where Columns $B_i$ represent the binary numbers of the obtained vector $Y_i$ [46].

**Table 3.** $Y_i$ value obtained in the proposed Binary-Addition-Tree (BAT) [20].

| *i* | $B_i$ | $Y_i$ | *i* | $B_i$ | $Y_i$ |
|---|---|---|---|---|---|
| 1 | 00000 | (0, 0, 0, 0, 0) | 17 | 10000 | (1, 0, 0, 0, 0) |
| 2 | 00001 | (0, 0, 0, 0, 1) | 18 | 10001 | (1, 0, 0, 0, 1) |
| 3 | 00010 | (0, 0, 0, 1, 0) | 19 | 10010 | (1, 0, 0, 1, 0) |
| 4 | 00011 | (0, 0, 0, 1, 1) | 20 | 10011 | (1, 0, 0, 1, 1) |
| 5 | 00100 | (0, 0, 1, 0, 0) | 21 | 10100 | (1, 0, 1, 0, 0) |
| 6 | 00101 | (0, 0, 1, 0, 1) | 22 | 10101 | (1, 0, 1, 0, 1) |
| 7 | 00110 | (0, 0, 1, 1, 0) | 23 | 10110 | (1, 0, 1, 1, 0) |
| 8 | 00111 | (0, 0, 1, 1, 1) | 24 | 10111 | (1, 0, 1, 1, 1) |
| 9 | 01000 | (0, 1, 0, 0, 0) | 25 | 11000 | (1, 1, 0, 0, 0) |
| 10 | 01001 | (0, 1, 0, 0, 1) | 26 | 11001 | (1, 1, 0, 0, 1) |
| 11 | 01010 | (0, 1, 0, 1, 0) | 27 | 11010 | (1, 1, 0, 1, 0) |
| 12 | 01011 | (0, 1, 0, 1, 1) | 28 | 11011 | (1, 1, 0, 1, 1) |
| 13 | 01100 | (0, 1, 1, 0, 0) | 29 | 11100 | (1, 1, 1, 0, 0) |
| 14 | 01101 | (0, 1, 1, 0, 1) | 30 | 11101 | (1, 1, 1, 0, 1) |
| 15 | 01110 | (0, 1, 1, 1, 0) | 31 | 11110 | (1, 1, 1, 1, 0) |
| 16 | 01111 | (0, 1, 1, 1, 1) | 32 | 11111 | (1, 1, 1, 1, 1) |

The value of each coordinate is either 0 or 1 in the binary-state network, with five coordinates in each vector. Hence, there are $2^5 = 32$ different vectors. Thus, all the possible state vectors are obtained without duplications in BAT [46].

## 3. Novel Dynamic BAT and States

The conventional path- and tree-like problems demonstrate well-defined destinations [32–34]. However, the propagation of the wildland fire problem considered in this study, its final destinations, and the number of destinations is unknown. Hence, it is difficult to estimate its possible occurrences in comparison with the conventional path and tree-like problems. The states and focus of the proposed algorithm are discussed in this section.

### 3.1. States and Maximum-State

A state of a node, i.e., node $i \in V$, is a node subset in $V(i)$, and these nodes in a state are all possible wildfires via node $i$. Hence, a state represents a possible wildfire that spreads from node $i$. A state has the following two properties:

1. the state of node $i$ can be an empty node subset because it is possible that no area is affected by the wildfire in node $i$,
2. these nodes in each state are subsets of $V(i)$ because the neighbor areas can face a wildfire spread from node $i$.

Hence, the possible state of each node, i.e., node $i$, is a node subset in $V(i)$ representing the possible areas of the wildfire spread during the same period. For example, in Figure 1, the states of node 0 are $\emptyset$ {5}, {7}, and {5, 7}, which are node subsets in $V(0)$. These states denote that a wildfire is possible in node 0 and does not spread, spread to node 5 but does not spread to node 7, spread to node 7 but does not spread to node 5, and spread to nodes 5 and 7, respectively.

The maximum-state is the state with the maximal number of nodes in $V(i)$, i.e., $V(i)$ is the maximum-state and the number of states is equal to the number of all possible node subsets of $V(i)$. The degree, adjacent node subset, and the number of combinations denoted by the notation $C(i)$ for node $i$ in Figure 1 are listed in Table 4 $C(i) = 2^{|Deg(i)|}$ and the maximum-state label of node $i$ is $2^{|Deg(i)|} - 1$, where $|\bullet|$ is the number of elements in set $\bullet$.

**Table 4.** $Deg(i)$, $V(i)$, $C(i)$, and $PR(i)$ values of node $i$.

| $i$ | $Deg(i)$ | $V(i)$ | $C(i)$ | $PR(i)$ | $PR(V(i))$ | $1 - PR(V(i))$ |
|---|---|---|---|---|---|---|
| 0 | 2 | {5, 7} | 4 | 0.074066 | 0.357809 | 0.642191 |
| 1 | 3 | {5, 6, 7} | 8 | 0.101160 | 0.549056 | 0.450944 |
| 2 | 2 | {6, 7} | 4 | 0.073398 | 0.357809 | 0.642191 |
| 3 | 3 | {5, 6, 7} | 8 | 0.101160 | 0.549056 | 0.450944 |
| 4 | 3 | {5, 6, 7} | 8 | 0.101160 | 0.549056 | 0.450944 |
| 5 | 6 | {0, 1, 2, 3, 4, 6} | 64 | 0.194502 | 0.642191 | 0.357809 |
| 6 | 6 | {1, 2, 3, 4, 5, 7} | 64 | 0.191247 | 0.734687 | 0.265313 |
| 7 | 5 | {0, 1, 3, 4, 6} | 32 | 0.163307 | 0.568793 | 0.431207 |

### 3.2. State Labels

Let us assume that $S_k(i)$ is the state corresponding to the $k$th state label of node $i$ for $i \in V$ and $k = 0, 1, \ldots, 2^{|Deg(i)|} - 1$. For example, from Table 3, $S_0(0) = \emptyset$, $S_1(1) = \{5\}$, $S_3(2) = \{6, 7\}$, $S_7(3) = \{5, 6, 7\}$, $S_2(4) = \{6\}$, $S_4(5) = \{0, 1\}$, $S_3(6) = \{3\}$, and $S_6(7) = \{1, 3\}$, respectively.

Each state has a unique state label. The first eight combinations of node $i = 0, 1, \ldots, 7$ in Figure 1 based on $V(i)$ are listed in Table 5, where the first column and first row denote the node label and state label, respectively.

**Table 5.** The first eight combinations of node $i$ based on $V(i)$ and Rules 1–3.

| Node \ State | 0 | 1 | 2 | 3 | 4 | 5 | 6 | 7 |
|---|---|---|---|---|---|---|---|---|
| 0 | $\emptyset$ | {5} | {7} | {5, 7} | | | | |
| 1 | $\emptyset$ | {5} | {6} | {7} | {5, 6} | {5, 7} | {6, 7} | {5, 6, 7} |
| 2 | $\emptyset$ | {5} | {6} | {6, 7} | | | | |
| 3 | $\emptyset$ | {5} | {6} | {7} | {5, 6} | {5, 7} | {6, 7} | {5, 6, 7} |
| 4 | $\emptyset$ | {5} | {6} | {7} | {5, 6} | {5, 7} | {6, 7} | {5, 6, 7} |
| 5 | $\emptyset$ | {0} | {1} | {2} | {0, 1} | {0, 2} | {1, 2} | {0, 1, 2} |
| 6 | $\emptyset$ | {1} | {2} | {3} | {1, 2} | {1, 3} | {2, 3} | {1, 2, 3} |
| 7 | $\emptyset$ | {0} | {1} | {3} | {0, 1} | {0, 3} | {1, 3} | {0, 1, 3} |

The state label is a number denoting a state's location among all the related states of each node. A state label is called a decimal state label or binary state label if it is represented by a decimal number or binary number, respectively. To recognize the state represented by a state label without losing any states, the notation used in BAT is adapted here to represent the state and state label of a node, i.e., $i \in V$, based on the following rules:

1. the number of digits is equal to $Deg(i)$ in the binary state label;
2. the nodes in the states of $i$ are arranged in the decreasing order of their node labels in $V(i)$;
3. the $i$th digit in the binary state label is equal to 0 or 1 if the $i$th node is included or excluded in the state, respectively.

The state label is a decimal state label if there is no further clarification. For example, in Figure 1, the degree of node 0 is 2, i.e., $Deg(0) = 2$, and its adjacent node subset is {7, 5}, i.e., $V(0) = \{7, 5\}$ (listed in the order of decreasing node labels by Rule 2). From Rules 1 and 3, these states $\emptyset$, {5}, {7}, and {5, 7} can be represented in the binary numbers 00, 01, 10, and 11, respectively, where the first/second digit in the binary numbers represents the state of node 7/5 based on the node listed order in $V(i)$. The corresponding decimal numbers of these binary numbers are state labels, i.e., state labels 0, 1, 2, and 3 are node subsets $\emptyset$, {5}, {7}, and {5, 7} because 0 = 00, 1 = 01, 2 = 10, and 3 = 11, respectively.

### 3.3. State Vectors

A state vector is a vector recording the sequence of the wildfire spread and its corresponding coordinates record the state labels of possible states of related nodes. In this study, the number of

coordinates in a state vector depends on the situation and is not constant. Furthermore, the nodes that are included in the vector and the coordinates representing these nodes are dependent on each other.

To identify the node with a state in the state vector, each coordinate is denoted by a two-dimensional tuple in a ratio such that the numerator is the current state label of the node shown in the denominator, i.e., 7/5 implies that node 5 is in the state (label) 7. For example, $X = (1/3, 1/5, 0/0)$ is a state vector including nodes 3, 5, and 0 with state labels 1, 1, and 0, respectively; and $X* = (4/3, 1/5, 0/6)$ is another state vector including nodes 3, 5, and 6 with state labels 4, 1, and 0, respectively.

An important advantage is that the origin and destination nodes of the wildfire spread listed in the state vector can be easily identified. For example, from $X = (1/3, 1/5, 0/0)$, we can identify that the wildfire is spread from nodes 3 to 5, then from nodes 5 to 0, and stops in node 0 because $S_1(3) = \{5\}$, $S_1(5) = \{0\}$, and $S_0(0) = \emptyset$ from $(1/3, 1/5, 0/0)$. Similarly, from $X* = (4/3, 1/5, 0/6)$, the wildfire starts from node 3, spreads to nodes 5 and 6 at the same time, and then spreads from node 5 to 0 and does not spread from node 6.

### 3.4. State Pagerank, State Probability, and Maximum-State Pagerank

In the Barabási-Albert model [45] discussed in Section 2.2, the preferential attachment in emphasizes that the probability of a node obtaining a connection is proportional to its node degree. Hence, here, we define the occurrence spread probability of possible states, called the state probability.

After obtaining the PageRank values [44] of all nodes using the algorithm presented in Section 2.3 and the possible states of each node provided in Sections 3.1–3.3, the next step is to calculate the state PageRank and state probability.

The state PageRank and state probability represent the PageRank value and occurrence probability of the related state, respectively. We assume that $PR(S_k(i))$ is the state PageRank of state $S_k(i)$ and $Pr(S_k(i))$ is the probability to obtain $S_k(i)$, where node $i \in V$ and $k = 0, 1, \ldots, 2^{|Deg(i)|}-1$. $PR(S_k(i))$ is calculated based on the PageRank of all nodes in the corresponding state of node $i$ in terms of the probability of a node obtaining a connection proportional to its node degree, i.e.,

$$PR(S_k(i)) = \sum_{|S_j(i)|=1 \text{ and } S_j(i) \subseteq S_k(i)} PR(S_j(i)). \tag{9}$$

Because the summation of the probability of all possible states is equal to one, we can normalize the related PageRank values as shown in the following equation:

$$Pr(S_k(i)) = \frac{PR(S_k(i))}{\sum_{i=0}^{2^{|Deg(i)|}-1} PR(S_k(i))} \tag{10}$$

where

$$PR(S_0(v)) = 0.5 \times Min\{W(S_k(i)) \mid \text{for } |S_k(i)| = 1 \text{ and } k = 0, 1, \ldots, 2^{|Deg(i)|} - 1\}. \tag{11}$$

From the preferential attachment in the Barabási-Albert model [18], a newly added node is free to connect to any node in the network; however, the probability of connecting to a degree-two node is twice that of connecting to a degree-one node. Hence, we have Equations (9)–(11).

For example, for node 0 in Figure 1, the PageRank values of states 0–3 are 0.0816538, 0.194502, 0.163308, and 0.357809, respectively, where the PageRank value of state 0 is set to be the smallest among the states with one single node, i.e., binary states 01 and 10 (decimal states 1 and 2). From the PageRank values, we have the occurrence probabilities of states 0–3 of node 0:

$$0.10241600 = 0.0816538/(0816538 + 0.194502 + 0.163308 + 0.357809) \tag{12}$$

$$0.24395900 = 0.194502/(0.0816538 + 0.194502 + 0.163308 + 0.357809) \tag{13}$$

$$0.20483300 = 0.163308/(0.0816538 + 0.194502 + 0.163308 + 0.357809) \tag{14}$$

$$0.44879200 = 0.357809/(0.0816538 + 0.194502 + 0.163308 + 0.357809). \tag{15}$$

The PageRank values and probabilities of the first eight states of each node are listed in Tables 6 and 7.

**Table 6.** PageRank values of the first eight states of each node.

| $i$ | 0 | 1 | 2 | 3 | 4 | 5 | 6 | 7 |
|---|---|---|---|---|---|---|---|---|
| 0 | 0.0816538 | 0.194502 | 0.163308 | 0.357809 | | | | |
| 1 | 0.0816538 | 0.194502 | 0.191247 | 0.385748 | 0.163308 | 0.357809 | 0.354554 | 0.549056 |
| 2 | 0.0956233 | 0.194502 | 0.191247 | 0.385748 | | | | |
| 3 | 0.0816538 | 0.194502 | 0.191247 | 0.385748 | 0.163308 | 0.357809 | 0.354554 | 0.549056 |
| 4 | 0.0816538 | 0.194502 | 0.191247 | 0.385748 | 0.163308 | 0.357809 | 0.354554 | 0.549056 |
| 5 | 0.0366988 | 0.0740667 | 0.10116 | 0.175227 | 0.0733977 | 0.147464 | 0.174558 | 0.248624 |
| 6 | 0.0366988 | 0.10116 | 0.0733977 | 0.174558 | 0.10116 | 0.20232 | 0.174558 | 0.275718 |
| 7 | 0.0370333 | 0.0740667 | 0.10116 | 0.175227 | 0.10116 | 0.175227 | 0.20232 | 0.276387 |

**Table 7.** Probabilities of the first 8 combinations of each node.

| $i$ | 0 | 1 | 2 | 3 | 4 | 5 | 6 | 7 |
|---|---|---|---|---|---|---|---|---|
| 0 | 0.10241600 | 0.24395900 | 0.20483300 | 0.44879200 | | | | |
| 1 | 0.03584640 | 0.08538720 | 0.08395830 | 0.16934600 | 0.07169290 | 0.15708000 | 0.15565100 | 0.24103800 |
| 2 | 0.11027700 | 0.22430800 | 0.22055400 | 0.44486200 | | | | |
| 3 | 0.03584640 | 0.08538720 | 0.08395830 | 0.16934600 | 0.07169290 | 0.15708000 | 0.15565100 | 0.24103800 |
| 4 | 0.03584640 | 0.08538720 | 0.08395830 | 0.16934600 | 0.07169290 | 0.15708000 | 0.15565100 | 0.24103800 |
| 5 | 0.00178264 | 0.00359777 | 0.00491382 | 0.00851160 | 0.00356528 | 0.00716305 | 0.00847910 | 0.01207690 |
| 6 | 0.00155856 | 0.00429615 | 0.00311711 | 0.00741326 | 0.00429615 | 0.00859229 | 0.00741326 | 0.01170940 |
| 7 | 0.00405280 | 0.00810559 | 0.01107060 | 0.01917620 | 0.01107060 | 0.01917620 | 0.02214120 | 0.03024680 |

### 3.5. The New BAT

In general, BAT can identify all possible state vectors (cases) that satisfy certain predefined conditions, e.g., a connected path, required amount/information flows, etc., between two specific node subsets in networks, i.e., one source-node to one target-node, one source-node to many target-nodes, many source-nodes to many target-nodes, and many source-nodes to one target-node [46]. However, to satisfy the real-time wildfire propagation conditions, the number of coordinates, the node representing a coordinate, and the state label for a node are unknown.

To meet the convenience of the conventional BAT for the practical problem discussed in this study, a novel dynamic BAT is proposed by implementing dynamic trees with dynamic lengths, flexible node orders, and different states.

We assume that the wildfire starts in the node (area) $i$, and we aim to identify all the possible scenarios where the wildfire spreads to at least $N_{area}$ areas and their corresponding probabilities. The procedure of the proposed BAT in determining the scenarios and probabilities are listed in Algorithm 3.

The number of coordinates in the state vectors may be different in the proposed BAT and the nodes connected may be present in the same state vectors as shown in Table 8.

For example, in Table 8, the initial state vector is $X = (1/3)$ if node 3 is the first identified area with a wildfire. The initial state vector always includes the first burned node and its state label starts from 1, as shown in line 3. Then, $X = (1/3, 0/5)$ due to $S_1(3) = \{5\}$ and node 3 does not reach its maximum-state. Furthermore, the other areas that are not the first identified area must begin with state 0, as shown in line 5.

If $N_{area} = 2$, we obtain a possible state vector because the wildfire in node 3 is propagated to a neighbor area, i.e., node 5, by line 6. If $N_{area} > 2$, we should increase the state of node 5 by one because the current state of node 5 is zero, i.e., it does not reach another node. The aforementioned analysis is based on line 3 and is similar to that of the conventional BAT.

---

**Algorithm 3** New-BAT Algorithm.

---

**Input** $G(V, E)$, $V(i)$, $S_k(i)$, and the node $s$ where the wildfire is first identified, for all $k = 0, 1, \ldots, 2^{|\text{Deg}(i)|} - 1$ and $i \in V$.

**Output** the probability $\Pr(s, N_{\text{area}})$ that the wildfire spreads to at least $N_{\text{area}}$ ($\geq 1$) areas (including the first area of wildfire identified).

1: Let $i = s$ be the node that the wildfire was first found, $l = 0$, the current sum of probabilities that the wildfire are spread out to at least $N_{\text{area}}$ areas $R = 0$, $T_l = \{i\}$, $T_{-1} = \emptyset$, the state label $X(i) = 1$.

2: **If** $X(i) = 2^{|\text{Deg}(i)|} - 1$, i.e., node $i$ reaches its the maximum-state **then go to** line 11.

3: Let $X(i) = X(i) + 1$ and $T^* = \{j \mid j \in [V(i) - T_l]\}$.

4: **If** $T^* = \emptyset$ **then go to** line 7.

5: Let $T_l = T_{l-1} \cup T^*$, $X(j) = 0$ for all $j \in T^*$, and $P_l = P_{(l-1)} \times \Pr(S_{X(i)}(i))$.

6: **If** $|S_l| \geq N_{\text{neighbor}}$, let $R = R + P_l$ **then go to** line 3.

7: Let new node $i$ be the node right after the current node $i$ in $T_l$ **then go to** line 4.

8: **If** there is no such node in line 7 **then go to** line 3.

9: Let $l = l - 1$, $i$ be the node right before the current node $i$ in $T_l$, and **go to** line 2. If there is no such node $i$, halt, and $\Pr(s, N_{\text{area}}) = R$ is the final probability that the wildfire can be spread out at least $N_{\text{area}}$ areas.

---

**Table 8.** Proposed BAT procedure to obtain the first state vector with five coordinates.

| $l$ | $i$ | $V(i)$ | Label [1] | Label [2] | $X(i)$ | $T^\#$ | $T_l = T_{l-1} \cup S^\#$ | $X$ | Probability * |
|-----|-----|--------|-----------|-----------|--------|--------|---------------------------|-----|---------------|
| 0 | 3 | $\{7, 6, 5\}$ | 1 | 001 | 1 | $\{5\}$ | $\{3, 5\}$ | (1/3) | $P_{3,\{5\}}$ |
| 1 | 5 | $\{6, 4, 3, 2, 1, 0\}$ | 0 | 000000 | 0 | $\emptyset$ | $\{3, 5\}$ | (1/3, 0/5) | $P_{3,\{5\}}P_{5,\emptyset}$ |
| 2 | 5 | $\{6, 4, 3, 2, 1, 0\}$ | 1 | 000001 | 1 | $\{5,7\}$ | $\{3, 5, 0, 7\}$ | (1/3, 1/5) | $P_{3,\{5\}}P_{5,\{0\}}$ |
| 3 | 0 | $\{7, 5\}$ | 0 | 00 | 0 | $\emptyset$ | $\{3, 5, 0, 7\}$ | (1/3, 1/5, 0/0) | $P_{3,\{5\}}P_{5,\{0\}}P_{0,\emptyset}$ |
| 4 | 7 | $\{6, 4, 3, 2, 1, 0\}$ | 0 | 000000 | 0 | $\emptyset$ | $\{3, 5, 0, 7\}$ | (1/3, 1/5, 0/0, 0/7) | $P_{3,\{5\}}P_{5,\{0\}}P_{0,\{7\}}P_{7,\emptyset}$ |
| 5 | 7 | $\{6, 4, 3, 2, 1, 0\}$ | 1 | 000001 | 1 | $\emptyset$ | $\{3, 5, 0, 7\}$ | (1/3, 1/5, 0/0, 1/7) | $P_{3,\{5\}}P_{5,\{0\}}P_{0,\{7\}}P_{7,\{1\}}$ |

[1] decimal labels [2] binary state labels * $p_{i,I} = \Pr(S_j(i)) = \Pr(X(i))$, where node subset $I$ is the $j$th state of node $I$ and $X(i) = j$.

The proposed BAT procedure to obtain the first state vector with five coordinates, i.e., where the number of wildfire areas $N_{\text{area}}$ is at least five, is provided and listed in Table 8.

### 3.6. Overall Procedure of the Proposed Method

The overall pseudo-code for the proposed method, based on the New BAT in Section 3.5 along with the novel state concepts discussed in Sections 3.1–3.4, in estimating the number of the wildfire propagation areas is presented in Algorithm 4.

---

**Algorithm 4** The overall procedure of we proposed methof.

---

**Input** The map including all areas we are interested in estimating the wildfire propagation

**Output** The total probabilities of the wildfire propagation areas.

1: Build the network model for the area map by letting the city/down/important area to be node and links to connect these areas that are neighbor to each other.

2: Generate and normalize the adjacency matrix based on each column.

3: Calculate the PageRank values for each node based on the degree of nodes as shown in Equation (4).

4: Find out all node subsets of each node as discussed in Section 3.1.

5: Calculate the state PageRank values (including the maximum-state PageRank values) and state probabilities based on Section 3.4.

6: Use the maximum-state PageRank values to decide which areas need to protect in the beginning, middle, and final stages of the wildfire propagation.

7: Use the proposed the New BAT provided in Section 3.5 to calculate $\Pr(i, N_{\text{area}})$ for all nodes $i$ and $N_{\text{area}} = 1, 2, \ldots, |V|$.

## 4. Experimental Results

The scale-free network illustrated in Figure 1 is used to validate the proposed algorithm in estimating the occurrence probability of the wildfire propagation areas, i.e., $Pr(i, N_{area})$ for $i = 0, 1,$ ..., $(n-1)$ and $N_{area} = 1, 2, ..., |V| = n$ without loss generality. In the experiment, we consider all occurrence probabilities for $N_{area} = 1, 2, ..., 8$ by assuming that the ignition area begins in nodes 0, 1, ..., 7, respectively. Hence, there are 64 tests included.

The proposed algorithm was coded in Python, and all 64 tests were run on Spyder, and performed on the Notebook on Windows 10 with an Intel Core i7-8650U CPU at 1.90 GHz and 2.11 GHz with 16 GB RAM.

Let $R_1 = R_{0.85}$ and $R_2 = R_{0.15}$ be the related probability for d = 0.85 and 0.15, respectively. The experimental results are listed in Tables 9 and 10, illustrated in Figures 2–6 which are analyzed in the following subsections.

**Table 9.** *Deg(i)*, *V(i)*, *Comb(i)*, and *PR(i)* values of node *i*.

| *i* | $N_{area}$ | Vector Number | Runtime | $R_1 = R_{0.85}$ | $R_2 = R_{0.15}$ | $R_1$-$R_2$ |
|---|---|---|---|---|---|---|
| 0 | 1 | 1 | 0.000 | 1.0000 | 1.0000 | 0.0000 |
|   | 2 | 5 | 0.000 | 0.8976 | 0.8919 | 0.0057 |
|   | 3 | 191 | 0.003 | 0.8938 | 0.8864 | 0.0074 |
|   | 4 | 1106 | 0.011 | 0.8895 | 0.8811 | 0.0084 |
|   | 5 | 18036 | 0.377 | 0.8863 | 0.8776 | 0.0087 |
|   | 6 | 356664 | 6.448 | 0.8776 | 0.8689 | 0.0087 |
|   | 7 | 7510836 | 84.470 | 0.8283 | 0.8253 | 0.0031 |
|   | 8 | 87742588 | 1131.378 | 0.5518 | 0.5654 | −0.0136 |
| 1 | 1 | 1 | 0.000 | 1.0000 | 1.0000 | 0.0000 |
|   | 2 | 13 | 0.000 | 0.9642 | 0.9612 | 0.0030 |
|   | 3 | 324 | 0.007 | 0.9620 | 0.9586 | 0.0034 |
|   | 4 | 2607 | 0.022 | 0.9586 | 0.9547 | 0.0040 |
|   | 5 | 36276 | 0.324 | 0.9524 | 0.9487 | 0.0038 |
|   | 6 | 773828 | 6.779 | 0.9349 | 0.9325 | 0.0025 |
|   | 7 | 13814130 | 152.826 | 0.8530 | 0.8617 | −0.0087 |
|   | 8 | 122668428 | 1591.609 | 0.5144 | 0.5412 | −0.0268 |
| 2 | 1 | 1 | 0.000 | 1.0000 | 1.0000 | 0.0000 |
|   | 2 | 5 | 0.000 | 0.8897 | 0.8904 | −0.0007 |
|   | 3 | 255 | 0.016 | 0.8875 | 0.8872 | 0.0003 |
|   | 4 | 1654 | 0.031 | 0.8840 | 0.8832 | 0.0008 |
|   | 5 | 21394 | 0.527 | 0.8782 | 0.8776 | 0.0005 |
|   | 6 | 415994 | 8.519 | 0.8647 | 0.8649 | −0.0003 |
|   | 7 | 8604502 | 117.473 | 0.8158 | 0.8213 | −0.0055 |
|   | 8 | 95178046 | 1374.070 | 0.5484 | 0.5675 | −0.0191 |
| 3 | 1 | 1 | 0.000 | 1.0000 | 1.0000 | 0.0000 |
|   | 2 | 13 | 0.000 | 0.9642 | 0.9612 | 0.0030 |
|   | 3 | 324 | 0.000 | 0.9620 | 0.9586 | 0.0034 |
|   | 4 | 2607 | 0.031 | 0.9586 | 0.9547 | 0.0040 |
|   | 5 | 36212 | 0.347 | 0.9524 | 0.9487 | 0.0038 |
|   | 6 | 771940 | 7.105 | 0.9349 | 0.9325 | 0.0025 |
|   | 7 | 13797962 | 167.760 | 0.8530 | 0.8617 | −0.0087 |
|   | 8 | 122821892 | 3518.926 | 0.5144 | 0.5412 | −0.0268 |
| 4 | 1 | 1 | 0.000 | 1.0000 | 1.0000 | 0.0000 |
|   | 2 | 13 | 0.000 | 0.9642 | 0.9612 | 0.0030 |
|   | 3 | 324 | 0.004 | 0.9620 | 0.9586 | 0.0034 |
|   | 4 | 2607 | 0.026 | 0.9586 | 0.9547 | 0.0040 |
|   | 5 | 36180 | 0.346 | 0.9524 | 0.9487 | 0.0038 |
|   | 6 | 771732 | 7.075 | 0.9349 | 0.9325 | 0.0025 |
|   | 7 | 13791562 | 156.802 | 0.8530 | 0.8617 | −0.0087 |
|   | 8 | 123244356 | 1742.573 | 0.5144 | 0.5412 | −0.0268 |

**Table 9.** *Cont.*

| $i$ | $N_{area}$ | Vector Number | Runtime | $R_1 = R_{0.85}$ | $R_2 = R_{0.15}$ | $R_1$-$R_2$ |
|---|---|---|---|---|---|---|
| 5 | 1 | 1 | 0.000 | 1.0000 | 1.0000 | 0.0000 |
|   | 2 | 252 | 0.003 | 0.9949 | 0.9963 | −0.0014 |
|   | 3 | 594 | 0.006 | 0.9865 | 0.9892 | −0.0027 |
|   | 4 | 8664 | 0.075 | 0.9798 | 0.9834 | −0.0036 |
|   | 5 | 119914 | 1.082 | 0.9696 | 0.9753 | −0.0057 |
|   | 6 | 2050424 | 18.796 | 0.9426 | 0.9540 | −0.0113 |
|   | 7 | 26321044 | 339.085 | 0.8306 | 0.8604 | −0.0298 |
|   | 8 | 183961358 | 2505.281 | 0.4628 | 0.5074 | −0.0446 |
| 6 | 1 | 1 | 0.000 | 1.0000 | 1.0000 | 0.0000 |
|   | 2 | 125 | 0.002 | 0.9984 | 0.9976 | 0.0008 |
|   | 3 | 355 | 0.005 | 0.9956 | 0.9942 | 0.0014 |
|   | 4 | 5216 | 0.077 | 0.9916 | 0.9904 | 0.0012 |
|   | 5 | 70006 | 1.172 | 0.9815 | 0.9811 | 0.0004 |
|   | 6 | 1153321 | 18.114 | 0.9573 | 0.9596 | −0.0023 |
|   | 7 | 14700763 | 178.070 | 0.8552 | 0.8706 | −0.0154 |
|   | 8 | 120803944 | 1547.187 | 0.4911 | 0.5204 | −0.0292 |
| 7 | 1 | 1 | 0.000 | 1.0000 | 1.0000 | 0.0000 |
|   | 2 | 124 | 0.002 | 0.9900 | 0.9918 | −0.0018 |
|   | 3 | 456 | 0.007 | 0.9787 | 0.9810 | −0.0022 |
|   | 4 | 7998 | 0.160 | 0.9725 | 0.9753 | −0.0028 |
|   | 5 | 130986 | 1.845 | 0.9650 | 0.9689 | −0.0039 |
|   | 6 | 2571160 | 35.229 | 0.9452 | 0.9528 | −0.0077 |
|   | 7 | 29212442 | 461.061 | 0.8386 | 0.8641 | −0.0255 |
|   | 8 | 179307046 | 2957.642 | 0.4680 | 0.5105 | −0.0425 |

**Table 10.** Node ranks based on Table 9 and PageRank values.

| $i$ \ $N_{area}$ | 1 | 2 | 3 | 4 | 5 | 6 | 7 | 8 | PR | 1-PR |
|---|---|---|---|---|---|---|---|---|---|---|
| **0** | 1 | 7 | 7 | 7 | 7 | 7 | 7 | 1 | 8 | 1 |
| **1** | 1 | 4 | 4 | 4 | 4 | 4 | 2 | 3 | 4 | 3 |
| **2** | 1 | 8 | 8 | 8 | 8 | 8 | 8 | 2 | 7 | 1 |
| **3** | 1 | 4 | 4 | 4 | 4 | 4 | 2 | 3 | 4 | 3 |
| **4** | 1 | 4 | 4 | 4 | 4 | 4 | 2 | 3 | 4 | 3 |
| **5** | 1 | 2 | 2 | 2 | 2 | 3 | 6 | 8 | 2 | 7 |
| **6** | **1** | **1** | **1** | **1** | **1** | **1** | **1** | 6 | **1** | 8 |
| **7** | 1 | 3 | 3 | 3 | 3 | 2 | 5 | 7 | 3 | 6 |

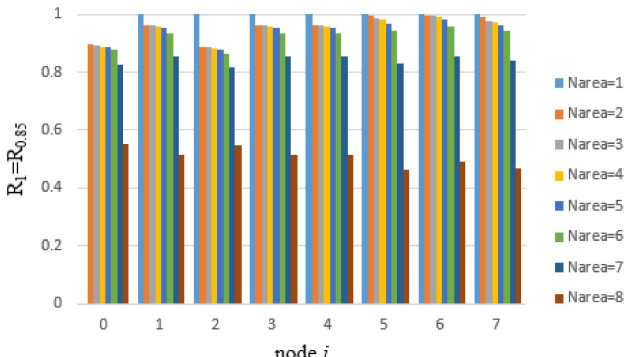

**Figure 2.** The probability of $R_1$ for $N_{area} = 1, 2, \ldots, 8$ in nodes $0, 1, \ldots, 7$.

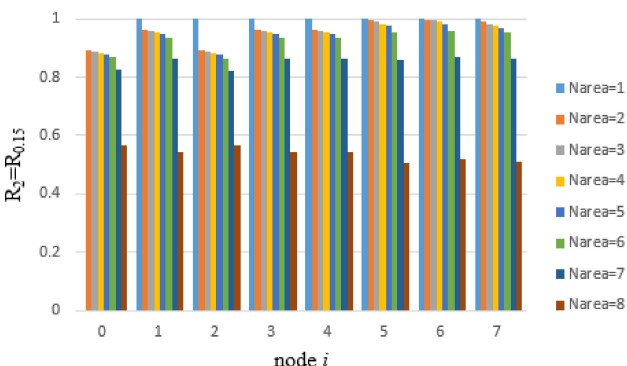

**Figure 3.** The probability of $R_2$ for $N_{area} = 1, 2, \ldots, 8$ in nodes $0, 1, \ldots, 7$.

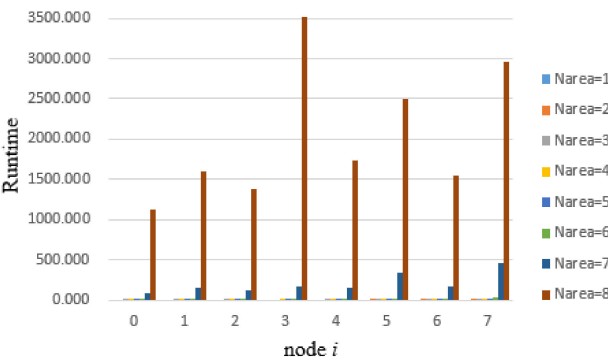

**Figure 4.** Runtime for $N_{area} = 1, 2, \ldots, 8$ in nodes $0, 1, \ldots, 7$.

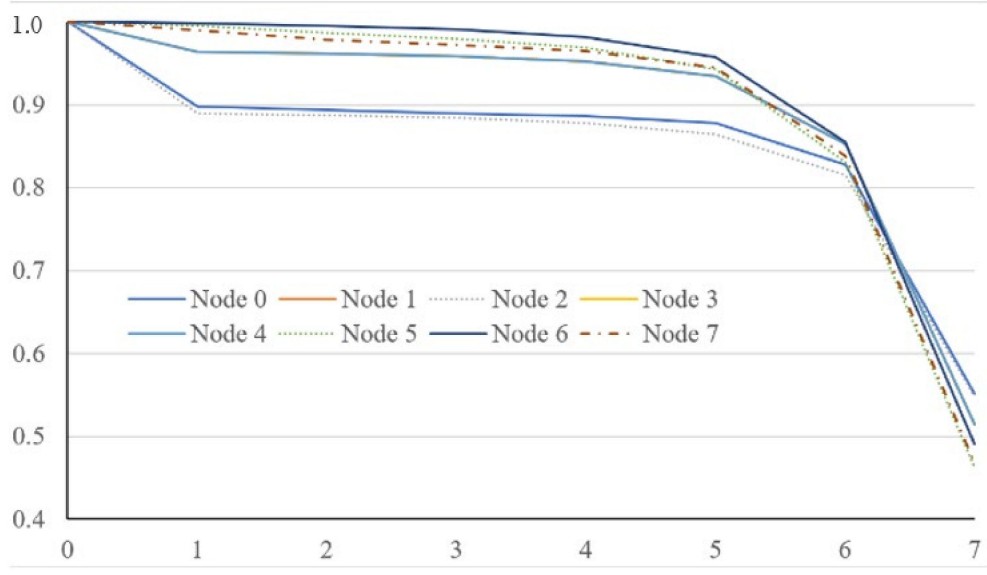

**Figure 5.** Maximum-state PageRank values.

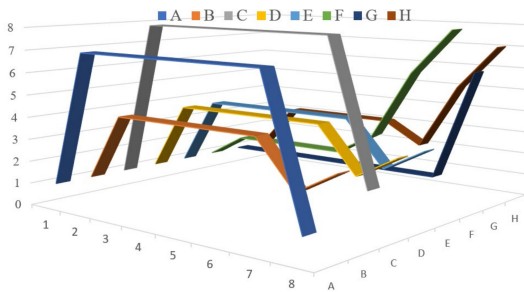

**Figure 6.** Node ranks based on maximum-state PageRank values.

### 4.1. General Results

The probability of one burned area, i.e., $N_{area} = 1$, is 100% because there is always an ignition area in the wildfire propagation, e.g., $Pr(i, N_{area} = 1) = 100\%$ for all node $i$ and $d = 0.85$ and $0.15$ as presented in Table 9.

From Table 9, the probability including $R_1 = R_{0.85}$ and $R_2 = R_{0.15}$ is decreased which is also further shown in Figures 2 and 3 for Figures representing the stages of how a network grows/densify, the number of state vectors is increased, and runtime is increased, which is also further shown in Figure 4 for Figures representing the stages of how a network grows/densifies with an increased $N_{area}$. In addition, from Figures 2 and 3, the probability including $R_1 = R_{0.85}$ and $R_2 = R_{0.15}$ is dramatically decreased when $N_{area}$ increases to 8. The aforementioned observation satisfies the real-life phenomena. Moreover, the number of state vectors and runtime are exponential for $N_{area}$ because the degree distribution in scale-free networks follows the power law.

### 4.2. Pr(i, N_{area}) and Deg(i)

From Table 9, the values of $R_{0.85}$–$R_{0.15}$ have a higher chance of being positive excepted for $i = 5$ and 7 and larger $N_{area}$. Hence, the related probabilities $d = 0.85$ (i.e., $R_{0.85}$) are often better than those of $d = 0.15$ (i.e., $R_{0.15}$).

Furthermore, from Table 9, if the adjacent node subsets of two different nodes are the same, e.g., $V(i) = V(j)$, we have $Pr(i, N_{area}) = Pr(j, N_{area})$ for all $N_{area}$. For example, for nodes 1, 3, and 4 in Figure 1, $V(1) = V(3) = V(4)$, and their probabilities of having at least $N_{area} = 1, 2, \ldots, 8$ burned areas are identical: 1.0000, 0.9642, 0.9620, 0.9586, 0.9524, 0.9349, 0.8530, and 0.5144 for $d = 0.85$, respectively, i.e., $Pr(i, N_{area}) = Pr(j, N_{area})$ for $i, j = 1, 3, 4$, and all values of $N_{area}$.

However, the total number of possible state vectors is different for a particular number of $N_{area}$. For example, for nodes 1, 3, and 4, $V(1) = V(3) = V(4)$, the number of vectors for nodes 1, 3, and 4 are different for $5 \le N_{area}$, e.g., the 36276, 36212, and 36180 correspond to nodes 1, 3, and 4 when $N_{area} = 5$.

It can be seen from Figure 2 for $d = 0.85$, the higher degree node seems to have a higher probability in lower $N_{area}$ but lower probability in higher $N_{area}$. For example, nodes 5 and 6 exhibit a degree of 6, which is the highest among all the nodes in Figure 1. $Pr(i = 6, N_{area})$ is always the highest among all other nodes for $N_{area} = 1, 2, \ldots, 7$ areas, but $Pr(i = 6, N_{area} = 8)$ ranks third from last. Similarly, $Pr(i = 5, N_{area})$ is less than $Pr(i = 6, N_{area})$ and its $Pr(i = 5, N_{area} = 8)$ is the least. On the contrary, $Pr(i = 2, N_{area})$ is always the least for $N_{area} = 1, 2, \ldots, 7$ but that of $Pr(i = 2, N_{area} = 8)$ is not the least.

Hence, it is necessary to extinguish wildfires in the areas with a higher degree in the beginning. However, the nodes with a lower degree for larger $N_{area}$ in the final stage of wildfire propagation require further attention.

### 4.3. Pr(i, N_{area}) and the Maximum-State PageRank

Moreover, the nodes with a better ranking in PageRank show the better ranking in the probability of smaller $N_{area}$; in contrast, these nodes with better ranking in $1 - $ (their PageRank values), i.e., the least PageRank values, show the better ranking in the probability of larger $N_{area}$. This is because

the higher degree of nodes always shows higher values of PageRank and better $Pr(i, N_{area})$ for smaller values of $N_{area}$.

For example, in Table 10, Figures 3, 5 and 6, A, B, ... , H denote nodes 0, 1, ... , 7 in Figure 1, node 6 exhibits the best maximum-state PageRank and its values of $Pr(6, N_{area})$ correspond well for $N_{area} = 1, 2, ... , 7$; node 1 exhibits the least maximum-state PageRank values and its $Pr(0, N_{area})$ correspond well for $N_{area} = 8$.

Hence, it is recommended to protect the areas with better maximum-state PageRank values at the beginning of the wildfire propagation and those with the least maximum-state PageRank values later. The observations further confirm the conclusion in Section 4.2.

### 4.4. Maximum-State PageRank

The proposed new BAT can calculate the correct value of $Pr(i, N_{area})$ for all nodes $i$ and $N_{area}$, i.e., the probability of the number of wildfire propagation areas, as presented in Table 9. However, the node degree distribution follows the power law, and the computation burden is increased following the power law based on the size of the scale-free network and the value of $N_{area}$. The aforementioned obstacle may overcome the requirement for a high-performance computer or GPU for a larger-size scale-free network.

The aforementioned analysis shows that the number of wildfire propagation areas is highly dependent on the proposed concept: maximum-state PageRank is the summation of all PageRank values in the maximum state of the related node, described in Sections 4.2 and 4.3.

Hence, in the aforementioned procedure, the computation burden may be avoided if we wish to identify the area that requires protection to reduce the wildfire propagation, i.e., we can consider $PR_{max}(i)$ if $Pr(i, N_{area})$ is not required.

### 5. Conclusions

In this study, our goal is to develop a novel, easily programable method to calculate the probability of the number of wildfire propagation areas.

First, we identify the possible wildfire landscapes from a scale-free network. These landscapes are modeled as a scale-free network generated from the Barabási-Albert model. Then, the PageRank algorithm is employed to calculate the distribution probability of each node. The probability of wildfire propagation areas is calculated based on the identified spread states that provide all possible states of each node and a new BAT to intelligently determine all scenarios of wildfire landscapes.

From the computational experiments conducted in 64 tests, the proposed BAT demonstrates the capability of systematically calculating the possible area numbers in wildfire propagation.

The proposed methodology presents analytical results. The proposed novel maximum-state PageRank values used in the algorithm can be separated to quickly determine the nodes (area) that reduce the probability of the wildfire propagation throughout, including at the beginning, middle, and end of the wildfire propagation. Without imposing a computational burden or affecting the final analytical results, the proposed novel maximum-state PageRank serves as a reliable and efficient tool in predicting wildfire propagation areas.

These results indicate that the proposed BAT, including the concept of the maximum-state PageRank, can be extended to large-scale problems. In the future, we will seek cooperation opportunities among the results of this study, the government, and private companies to apply the results of this study to practical cases. Besides which, the Climate conditions as factors in the equations and random values will also be incorporated into the research model in future studies.

**Author Contributions:** Conceptualization, W.-C.Y.; methodology, W.-C.Y.; software, W.-C.Y.; validation, W.-C.Y. and C.-C.K.; formal analysis, W.-C.Y. and C.-C.K.; investigation, W.-C.Y.; resources, W.-C.Y.; data curation, W.-C.Y.; writing—original draft preparation, W.-C.Y. and C.-C.K.; writing—review and editing, W.-C.Y. and C.-C.K.; visualization, W.-C.Y.; supervision, W.-C.Y.; project administration, W.-C.Y.; funding acquisition, W.-C.Y. All authors have read and agreed to the published version of the manuscript.



**Funding:** This research was funded by the Ministry of Science and Technology, R.O.C., grant number MOST 102-2221-E-007-086-MY3 and MOST 104-2221-E-007-061-MY3.

**Acknowledgments:** The author wishes to thank the anonymous editor and the referees for their constructive comments and recommendations, which significantly improved this article. This research was supported in part by the Ministry of Science and Technology, R.O.C. under grant MOST 102-2221-E-007-086-MY3 and MOST 104-2221-E-007-061-MY3. This article was once submitted to arXiv as a temporary submission that was just for reference and did not provide the copyright.

**Conflicts of Interest:** The authors declare no conflict of interest.

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
