# Peer review of "Predicting and Modeling Wildfire Propagation Areas with BAT and Maximum-State PageRank"

_applsci, doi:10.3390/app10238349_

Round 1

Reviewer 1 Report

Summary: Authors have proposed an algorithm based on BAT and maximum-State PageRank method to efficiently determine the wildfire propagation area. The topic is interesting, however needs to be improved in terms of editing. Following are the comments which could be addressed for further improvement of the overall quality of the manuscript.

(1) Please explain what is non-structure and non-prescribed refer to on page 1, line 28

(2) Page 1, line 30 is repeated.

(3) Briefly explain the main reasons and how the wildfire propagates in realtime

(4) Include some statistical information regarding the impact of the wildfire.

(5) Line 53-73 is repeated.

(6) What are advantages of using BAT and PageRank over other methods?

(7) How was dataset generated?

(8) How closely the data set resembles the actual scenario?

(9) Flame length has been used in equation 2, how about the propagation radius?

(10) Please discuss the computational complexity.

Author Response

Reviewer#1, Concern # 1:

1.Please explain what is non-structure and non-prescribed refer to on page 1, line28

Author response:  Thank you very much for the comment.

Author action: The definition of wildfire refers to the first reference (Firewise, 1998). Firewise (1998) defines wildfire as a non-structure fire  that occurs in the wildland , other than prescribed fire. In other words, wildfire is an unplanned, unwanted wildland fire including unauthorized human-caused fires, escaped wildland fire use events, escaped prescribed fire projects.

Reviewer#1, Concern # 2:

  1. Page 1, line 30 is repeated.

Author response:  Thank you very much for the comment.

Author action:  The article has been rechecked and the repeated paragraphs have been removed.

Reviewer#1, Concern # 3:

  1. Briefly explain the main reasons and how the wildfire propagates in real-time.

Author response:  Thank you very much for the comment.

Author action:  Viegas (1998) indicated there are number of factors involved the wildfire, and there are no general models yet available to perform this task with a sufficient degree of accuracy and reliability. There is a wide range of fire propagation models, ranging from the empirical to the physical (Rothermel, 1990; Andre et al, 1992). In this study, we demonstrated a scale-free model as Figure 1, which is more general application in principle. The hypothesis model based on the Barabási-Albert model which is arguably the most popular scale-free model in part B of Section II. The addition of the new nodes is in accordance with a probability, not a fixed number. The connection of the initial and the new nodes is also based on the Barabási-Albert model in part B of Section II. When the new node is added, previous links are not deleted, but their probability of adding a new node would be also based on its degree. In Figure1, the node labels are also arbitrarily assigned based on the Barabási-Albert model and they have not hierarchy. This model only considers the propagates of wildfires, more complexity of the phenomena involved and to the number and type of input data that would be considered. 

Reviewer#1, Concern # 4:

  1. Line 53-70 is repeated.

Author response:  Thank you very much for the comment.

Author action:  The article has been rechecked and the repeated paragraphs have been removed.

Reviewer#1, Concern # 5:

  1. What are advantages of using BAT and PageRank over other methods?

Author response:  Thank you very much for the comment.

Author action:  We propounded and proved that wildfire propagation is a scale-free network generated from the Barabási–Albert model. From Barabási and Albert (1999) showed that the PageRank distribution in the scale-free model satisfies the power law. Using BAT algorithm that employs binary addition for finding all the possible state vectors and the path-based layered-search algorithm for filtering out all the connected vectors is proposed for calculating the binary-state network reliability.

Reviewer#1, Concern # 6:

  1. How was dataset generated?

Author response:  Thank you very much for the comment.

Author action: The model and dataset was generated from the Barabási-Albert model.

Reviewer#1, Concern # 7:

  1. How closely the data set resembles the actual scenario?

Author response:  Thank you very much for the comment.

Author action:

There is no general model of wildfire propagation. Even weather forecasting models are constructed from years of data and related influence factors. This study proposed a scale-free model as a general model for wildfire propagation. It can be applied this model as a general model. We will consider more restrain factors into the wildfire propagation.

Reviewer#1, Concern # 8:

  1. Flame length has been used in equation 2, how about the propagation radius?

Author response:  Thank you very much for the comment.

Author action: In this study, we used flame length to verify the scale-free model. Therefore, the radius of propagation is not considered.

Reviewer#1, Concern # 9:

  1. Please discuss the computational complexity.

Author response:  Thank you very much for the comment.

Author action: The model computational complexity is Ο(n). The n means the complexity of the arc will grow in direct proportion to the size of the input data..

Reviewer 2 Report

In this paper, the author implemented a scale-free network model incorporating Binary-Addition-Tree (BAT) together with the PageRank. The issue of wildland fire is worldwide with very devasting consequences. An effective model that can predict the probability of fire propagation is paramount, especially one that can run faster than real-time. However, there are many critical issues in this article. It is recommended to undergo major revision.

Firstly, this paper is purely a proposal for a numerical modelling framework. There is a large section for illustrating of the methodology. There were no description of the wildland fire spread scenario being considered (if there is any) or any post-processing of the numerical results to convert into actual useful data for wildland fire operations.

Section 4 – Experimental Results section is very weak. Usually, validation (especially validation for wildland fire spread) is based on actual experimental data. In this article, the author claims the model was validated using ‘’scale-free network illustrated in Fig. 1’’. Validation of a numerical model using another generic model is flawed.

Therefore, the author need to significantly strengthen the validation and experimental results section. It is recommended to apply the model for a more realistic wildfire scenario. Wildland fire journal papers usually involve a actual real-life case study.

Lastly, there are many grammatical errors and typos in the article.

In the introduction, there are some contents that were repeated. For example, Line 63 - 73 – The majority of the content here is a complete repeat of previous paragraphs.

Author Response

Reviewer#2, Concern # 1:

  • There were no descriptions of the wildland fire spread scenario being considered (if there is any) or any post-processing of the numerical results to convert into actual useful data for wildland fire operations.
  • Validation (especially validation for wildland fire spread) is based on actual experimental data. Therefore, the author need to significantly strengthen the validation and experimental results section. It is recommended to apply the model for a more realistic wildfire scenario.

Author response:  Thank you very much for the comments, and we agree with your opinion.

Author action:  We demonstrated a scale-free model as a general model. There is no general model of wildfire propagation. Even weather forecasting models are constructed from years of data and related influence factors. This study proposed a scale-free model as a general model for wildfire propagation. It can be applied this model as a general model. We will consider more restrain factors into the wildfire propagation.

Reviewer#2, Concern # 3:

3- there are many grammatical errors and typos in the article.

Author response:  Thank you very much for the comments. We agree with this view and will aim to make changes accordingly.

Author action:  The article has been rechecked and the repeated paragraphs have been removed.

Reviewer#2, Concern # 4:

4- In the introduction, there are some contents that were repeated. For example, Line 63-73.

Author response:  Thank you very much for the comments. We agree with this view and will aim to make changes accordingly.

Author action:  The article has been rechecked and the repeated paragraphs have been removed.

Round 2

Reviewer 2 Report

Most of the comments have been addressed by the author. However, the English can be further improved with an overall check (i.e. particularly in Introduction, Abstract and Conclusion). It is recommended a final grammatical check and read through before publishing.